# The Efficacy of Er:YAG Laser-Activated Shock Wave-Enhanced Emission Photoacoustic Streaming Compared to Ultrasonically Activated Irrigation and Needle Irrigation in the Removal of Bioceramic Filling Remnants from Oval Root Canals—An Ex Vivo Study

**DOI:** 10.3390/bioengineering9120820

**Published:** 2022-12-19

**Authors:** Gabrijela Kapetanović Petričević, Marko Katić, Valentina Brzović Rajić, Ivica Anić, Ivona Bago

**Affiliations:** 1Department of Endodontics and Restorative Dental Medicine, School of Dental Medicine, University of Zagreb, 10000 Zagreb, Croatia; 2Department of Materials, Faculty of Mechanical Engineering and Naval Architecture, University of Zagreb, 10000 Zagreb, Croatia; 3Department of Dental Diseases, Dentistry Clinic, University Hospital Centre, 10000 Zagreb, Croatia

**Keywords:** retreatment, Er:YAG laser, active irrigation, oval root canal, filler

## Abstract

The removal of filling material is important for successful root canal retreatment. The aim of the study was to compare the efficiency of two activated irrigation techniques, the shock wave-enhanced emission photoacoustic streaming (SWEEPS) mode of the Er:YAG laser and ultrasonically activated irrigation (UAI) and a conventional syringe-needle technique (SNI), in the removal of bioceramic sealer/gutta-percha during conventional retreatment in oval root canals. The study sample consisted of distal root canals of 42 extracted human mandibular molars, which were prepared using a ProTaper Next system up to size 40/0.06 and filled with bioceramic sealer using a single-cone obturation technique. The teeth were then re-treated with a Reciproc Blue RB40 file and 3% sodium hypochlorite solution. The prepared teeth were randomly divided into three groups (*n* = 14 per group) and subjected to one of the three irrigation methods. Micro-CT scans were performed at different stages to assess the amount of filling material after each retreatment phase. The results of the study showed that all the tested irrigation techniques reduced a statistically significant amount of the remnant filling material at retreatment (*p* < 0.05), and there were no statistically significant differences in efficacy between the three methods. All tested techniques had similar efficacy in the removal of the remaining filling remnants.

## 1. Introduction

The main aim of root canal retreatment is the removal of the filling material to provide re-disinfection, re-instrumentation, and re-filling of the endodontic space, which are the main prerequisites for periapical healing [1]. Owing to the complex root canal anatomy, a significant amount of filling material remains on root canal dentin or in canal complexities (approximately 10% in the coronal third and 90% in the apical third of the root canal), causing limited penetration of irrigants and difficult re-cleaning of these areas [2,3,4]. Therefore, in recent years, the effectiveness of different instrumentation systems and irrigation techniques has been investigated to determine the best chemo-mechanical protocol for retreatment. Most of these studies concluded that it was almost impossible to completely remove the filling material from the root canal system [2]. Furthermore, the degree of removal depends not only on the instrumentation and irrigation technique used but also on the type of sealer and root canal anatomy [4,5,6,7]. Oval root canals are considered the most problematic for successful retreatment [2,8] because of the untouched flattened areas of the canal [9,10,11,12]. Recent studies reported the benefit of additional root canal instrumentation with XP-Endo Finisher to remove filling remnants [7].

The removal of filling remnants left after conventional chemo-mechanical retreatment can be improved by activated irrigation during the final irrigation protocol [4,5,13]. Ultrasonically activated irrigation (UAI) has been proven to be a very effective supplementary technique after the retreatment of roots initially filled with gutta-percha and epoxy resin-based or bioceramic-based materials [4,13,14]. However, in some studies, its effectiveness was compromised when used in complex canal anatomies or smaller root canals compared to instruments that adapt to the intracanal anatomy, such as the XP-endo Finisher [15,16] and laser-activated irrigation (LAI) [4,17,18]. In a recent study by Baumeier et al. [19], there were no differences between UAI, EndoActivator, and XP-endo Finisher R in additional cleaning of filling remnants from a flattened root canal. Regardless of the stated possible shortcomings, UAI is still considered a “gold standard” in activated root canal irrigation; hence, every new irrigation system has to be compared to it [20].

Recently, a new mode of erbium: yttrium aluminum garnet (Er:YAG) LAI has been launched called Shock Wave-Enhanced Emission Photoacoustic Streaming (SWEEPS^®^). Its efficacy is based on the delivery of pairs of ultrashort pulses (25 µs) with minimal energy levels (25 mJ) into an irrigant in the root canal [21]. This laser–liquid interaction creates a series of bubbles in the irrigant that is timed to appear such that secondary bubbles lead to the collapse of existing primary bubbles, resulting in very strong shock waves and photoacoustic streaming [22]. The SWEEPS mode has been shown to be very promising for the removal of smear layers and debris [23] from complex regions in the root canal and for root canal retreatment [17].

The majority of the published studies on the efficacy of retreatment were based on root canals filled with gutta-percha and epoxy resin-based sealers [15,16,17,18]. Recently, attention has been given to the evaluation of the retreatment of bioceramic cement (BC) used for root canal filling. These hydraulic cements are unique because of their interaction with phosphates in tissue fluids and the consequent formation of hydroxyapatite precipitates [6,24,25]. As a result, they show higher bond strength to root dentin than epoxy-resin-based sealers [26]. Studies on their re-treatability showed that none of the retreatment techniques completely removed gutta-percha/BC from the oval root canals [6]. Furthermore, bioceramic sealer (BioRoot RCS) can be easily removed as a zinc oxide-eugenol sealer [27]. According to a recent review by Arul et al. [6], further studies on the retrievability of bioceramic sealers are required.

The aim of this study was to evaluate the effectivity of the new Er:YAG LAI mode, SWEEPS^®^, in removing bioceramic filling remnants during the retreatment of oval root canals and also to compare it with the “golden standard” UAI and conventional syringe-needle irrigation (SNI). The null hypothesis was that there would be no difference in the reduction of root canal filling remnants among the tested irrigation protocols.

## 2. Materials and Methods

### 2.1. Samples Selection

The study protocol was approved by the Local Ethics Committee with reference number 05-PA-30-VI-4/2019. A power analysis was performed using the chi-squared test (a = 0.05 and b = 0.95), and a minimum of 10 canals were calculated as the sample size for each group.

A total of 42 oval distal root canals with straight curvature (<5%) were selected from a group of extracted human mandibular molars from patients aged between 22 and 42 years old and were used in the study [28]. Oval root canals were selected by means of a cone-beam computed tomography scan (Cranex 3DX; Soredex, Tuusula, Finland) using the following parameters: field of view, 595 (5.0 mm) mm; ENDO, 85 µm; 6.3 mA; 90 kV; 8.7 s; 450.3 mGycm^2^ and measured by criteria based of De Deus et al. [14]. The exclusion criteria were: previous root canal treatment, intracanal calcifications, root caries, external resorption, and internal resorption. The selected teeth were stored in 0.1% thymol solution at 4 °C before use. Inclusion criteria were: apical diameter of all the selected samples corresponding to an ISO size range of 15–20 (confirmed by #15 K-file and #20 K-file).

### 2.2. Preparation of Root Canals and Root Canal Filling

The same operator performed all steps of the root canal treatment. Traditional access openings with straight-line access to root canals were prepared using a water-cooled diamond fissure No. 016 (Komet, Rock Hill, SC, USA). Canal patency was confirmed using a 15/20 K-file (Dentsply Sirona Endodontics), according to the inclusion criteria. The working length (WL) was determined by subtracting 0.5 mm from the length obtained when the K-file instrument was visible at the apical foramen using loupes of 4.5× magnification. The apical foramen of all selected roots was sealed with hot glue and embedded in polyvinylsiloxane (Exaflex putty, GC) to prevent irrigation fluid flow through the apical foramen [4].

Root canals were prepared using the ProTaper Next (PTN) rotary system (Dentsply Sirona Endodontics) up to a size of 40/0.06 using the VDW Gold motor set in rotary mode (300 rpm, 270 NCm). During instrumentation, a total of 10 mL of 3% sodium hypochlorite (NaOCl) was used to irrigate the root canals using a 31G needle (SteriTips, DiaDent, Burnbay, BC, Canada). After chemo-mechanical instrumentation, the smear layer was removed by the conventional final irrigation protocol proceeded as follows: 2 mL of 3% NaOCl for 30 s, 2 mL of 15% ethylenediaminotetraacetic acid (EDTA) (Calsinase, Dettenhausen, Germany) for 1 min and 2 mL of 3% NaOCl for 30 s.

After instrumentation, the canals were dried using sterile PTN X4 paper points (ProTapex Next, Dentsply Sirona Endodontics, San Clemento, CA, USA).

### 2.3. Root Canal Filling

The root canals were filled with a dual-component bioceramic sealer (BioRoot RCS; Septodont, France) using a single-cone obturation technique. BioRoot RCS was prepared according to the manufacturer’s recommendations. Thereafter, a PTN X4 gutta-percha point was dipped into the sealer and inserted slowly into the canal up to the WL. The gutta-percha cone was cut at the canal orifice and vertically condensed using a size 4 hand cold plugger (Machtou plugger; VDW, Munich, Germany). After the root canal filling, access cavities were temporarily restored (Caviton; GC, Tokyo, Japan). All samples were stored at 37 °C and 100% relative humidity for 2 weeks.

### 2.4. Root Canal Retreatment

The samples were retreated using a reciprocating technique with the Reciproc Blue RB40 (40/0.06) file (VDW Dental, München, Germany) and the VDW Gold motor set at the Reciprocation All mode. During the retreatment procedure, each root canal was irrigated with 6 mL of 3% NaOCl solution. The instrument was advanced apically using an in-and-out pecking motion with an amplitude of approximately 3 mm according to the manufacturer’s instructions. After three pecks, the instrument was cleaned with sterile gauze, and the canal was irrigated with 3% NaOCl. Each RB40 instrument was used to retreat three canals. Retreatment was completed when each instrument reached the WL three consecutive times.

Root canals were then dried using sterile paper points (RECIPROC^®^ Paper Points R40) [29].

### 2.5. Experimental Final Irrigation Protocol

Following the retreatment procedure, the samples were randomly divided into three groups (*n* = 14 each) according to the experimental final irrigation technique.

Group 1. SWEEPS^®^

The root canals were irrigated with a total of 6 mL 3% NaOCl. The irrigant was delivered continuously using a 31 G side vented needle (SteriTips, DiaDent, Burnbay, BC, Canada) while the tip was placed stationary in the access cavity. Simultaneously, the irrigant was activated with an Er:YAG laser (LightWalker AT, Fotona, Ljubljana, Slovenia) using a radial endodontic laser tip (diameter 600 μm, 9 mm long) (Fotona, Ljubljana, Slovenia) for 90 s in total, divided into three cycles of 30 s each. The radial laser tip was placed stationary in the access cavity during activation. The laser parameters were as follows: Auto SWEEPS mode, pulse energy of 20 mJ, pulse repetition rate of 15 Hz, average power of 0.30 W, pulse duration of 25 µs, and peak power of 800 W.

After LAI was completed, the remaining irrigant was aspirated from the canal using a syringe and 31-G needle (Steri Tips, DiaDent, Burnbay, BC, Canada) [4].

Group 2. Ultrasonically activated irrigation

The samples were irrigated with a total of 6 mL of 3% NaOCl. The irrigant was delivered into the root canal using a 31G side vented needle (Steri Tips, DiaDent, Burnbay, BC, Canada) placed 1 mm to the WL. Each sample was continuously activated with an endodontic ultrasonic tip (Irri S, 25/25, VDW) using an endodontic ultrasonic source (VDW ULTRA; VDW, München, Germany). The ultrasonic tip was placed 3 mm from the WL. The duration of activation was 90 s, which was divided into three cycles of 30 s each. After UAI, the remaining irrigant was aspirated from the canal using a syringe and 31-G needle (Steri Tips, DiaDent, Burnbay, BC, Canada).

Group 3. Conventional syringe-needle irrigation

A 31G side vented needle (Steri Tips, DiaDent, Burnbay, BC, Canada) was used to irrigate each root canal with a total of 6 mL of 3% NaOCl, applied within 90 s. The syringe was placed 1 mm below the WL. The duration of the intracanal application was three cycles of 30 s each, that is, one cycle for each 2 mL of irrigation solution. After SNI was completed, the remaining irrigant was aspirated using a syringe and a needle of the same size.

### 2.6. Micro-Computed Tomography (μCT) Analysis

All samples were scanned three times using a μCT) device (XT H 225; Nikon, Tokyo, Japan): after root canal filling (Volume I), after root canal retreatment (Volume II), and after the experimental final irrigation protocol (Volume III).

The volume of filling material was measured using an industrial micro-CT (Nikon XT H 225) device with a target focal size of 0.7 μm and a 400 mm × 300 mm 14-bit flat panel detector with 127-μm pixel size. The samples were measured at 80 kV and 60 μA using 1600 projections at an exposure time of 1 s. The geometrical magnification was ~100, which yielded a structural resolution of 1.2 μm. A threshold algorithm was used to find out the volume of the filling material inside the root canal based on the grayscale value for the tooth as the base value. All the samples were scanned at the same position with the same radiation settings. During analysis, the filling material was treated as an inclusion in the base material (hard tooth tissue). The results are expressed as the percentage of filling material in the base material (hard tooth tissue).

### 2.7. Statistical Analyses

The decrease in the amount of filling material after each retreatment protocol was analyzed using the Kruskal–Wallis test. Intergroup analyses were performed using the Wilcoxon test. Statistical significance was set at *p* < 0.05. All statistical analyses were performed using IBM SPSS Statistics for Windows version 23.0 (www.spss.com (accessed on 10 September 2022)).

## 3. Results

The initial volume of filling material was similar in all groups (mean 6.08–6.95 mm^3^, Table 1) (*p* > 0.05).

The μCT three-dimensional models of teeth showed the remaining filling remnants after basic retreatment and after the final irrigation protocol in each group (Figure 1)

The conventional chemo-mechanical retreatment (Volume II compared to Volume I) and final irrigation protocol (Volume III compared to Volume II) contributed significantly to the removal of gutta-percha/BC filling material in all groups (*p* < 0.001). There were three samples without any filling remnants in the SWEEPS group and one sample in the UAI group.

There were no statistically significant differences in the reduction rate of the filling material between the groups in any retreatment phase (*p* > 0.05) (Table 1).

Table 1 shows the volume of the filling material and the reduction rate of the initial volume of the filling material (Volume I) in mm^3^, and the remaining volume of the material after the basic retreatment procedure (Volume II) and after the experimental irrigation protocol (Volume III) for each group. Table 1 also shows the reduction rate (%) of the filling material after the retreatment procedure (compared to Volume I) and after the final irrigation protocol (compared to Volume II) for each of the final irrigation protocols tested.

## 4. Discussion

The results from this study showed that conventional chemo-mechanical retreatment using reciprocating instrumentation technique without final irrigation protocol can easily remove significant amounts of gutta-percha with a bioceramic sealer (BC) from oval root canals with mean reduction rates range of 86.61–90.90%. The mean value of the remaining gutta-percha/BC ranged from 0.72 to 0.89 mm^3^. The high reduction rate of the filling material may be attributed to the high efficacy of the reciprocating instrumentation technique using the Reciproc Blue RB40/06, which has an S-shaped cross-section, sharp cutting edges, and large chip space feature designs that provide efficient mechanical retreatment [10]. High efficacy of Reciproc Blue instruments in the retreatment was already reported in previous studies [29]. The canals were instrumented to a size of 40/06 and filled using the single-cone technique. None of the samples were completely cleaned from the filling material. The same finding regarding the re-treatability of bioceramic sealers was reported in a recent review by Arul et al. [6]. Other studies have shown that it is impossible to completely remove BC from oval root canals using conventional retreatment techniques, as is also the case with epoxy-resin-based sealers [6,12,27,30]. Furthermore, previous studies did not find a difference in the re-treatability between BC and epoxy-resin-based sealers [5,31]. Although this was not the aim of this study, our results showed that BC combined with the gutta-percha point can easily be re-treated. Future studies should evaluate the re-treatability of pure BioRoot RCS without gutta-percha point. Eymirli et al. [32] reported that gutta-percha facilitated the removal of a calcium silicate-based sealer (Endosequence BC Sealer) and the re-establishment of the WL.

In this study, all three final irrigation protocols contributed to the removal of BC/gutta-percha filling remnants without significant differences between them; therefore, the null hypothesis of the study was confirmed. The UAI is a well-known activation technique, and its efficacy in retreatment has been proven in many previous studies [4,33]. SWEEPS is a new LAI technique that is based on Er:YAG laser irradiation and dual-pulse technology. Its efficacy has been evaluated in many recent studies at different stages of root canal treatment, including smear layer and debris removal [34,35], removal of organic material [36], and retreatment [4,17]. Although SWEEPS showed promising results, some studies show that it did not overcome the limitations of traditional UAI techniques [4,37,38]. A few recent studies have evaluated the efficacy of SWEEPS for retreatment. In a study by Bago et al. [17], SWEEPS^®^ was more successful than UAI in the additional removal of epoxy resin-based sealer and gutta-percha from curved root canals when NaOCl (time of irrigation 3 × 20 s) was used. The root canals were filled using the continuous-wave vertical compaction technique and warm injection back-filling technique. In a later study by the same research group [4], the SWEEPS technique combined with NaOCl (3 × 30 s) had similar efficacy as UAI, regardless of the laser tip used. Although the methodologies of the two studies were similar, in the first study [17], the smear layer was removed with EDTA and NaOCl before the application of the irrigation protocol. This step was not applied in the second study [4], which could explain the worse results in the UAI group. In a recent study by Angerame et al. [30], the SWEEPS technique in combination with EDTA, NaOCl, and distilled water showed better performance than UAI in the additional removal of BioRoot RCS/gutta-percha from intact single-rooted teeth after reciprocating retreatment. Based on these studies, the anatomy of the root canal could influence the performance of an additional activated irrigation protocol. Future studies should evaluate the removal of bioceramic sealers from curved root canals.

The results of this study showed that activated techniques, SWEEPS, UAI, and SNI, significantly removed the BioRoot RCS filling remnants from oval root canals after chemo-mechanical instrumentation using a single reciprocating file. All three techniques showed similar efficacy. However, it should be noted that SWEEPS was the only modality with the highest number of samples (3/14) to be completely rid of filling remnants. Therefore, the SWEEPS technique has a greater potential for root canal cleaning.

The limitation of this study was that since no analysis of the canal thirds was made, we cannot exclude the influence of the different canal regions on the effectiveness of the different final irrigation methods. It should also be noted that the results of this study were obtained for samples with a traditional access cavity; however, it is known that the type and size of the access cavity can influence the performance of Er:YAG LAI techniques, including SWEEPS [39]. Furthermore, the differences between groups, considering Volume III vs. Volume I, were not significant; however, they showed values very close to the level of significance (*p* = 0.067). Therefore, future studies should include bigger sample sizes in order to check if the difference remains the same.

## 5. Conclusions

The SWEEPS mode, UAI, and SNI, in combination with NaOCl, contributed to the additional removal of BioRoot RCS/gutta-percha from oval root canals after reciprocating instrumentation. The SWEEPS mode has great potential for the complete removal of filling remnants during the retreatment.

## Figures and Tables

**Figure 1 bioengineering-09-00820-f001:**
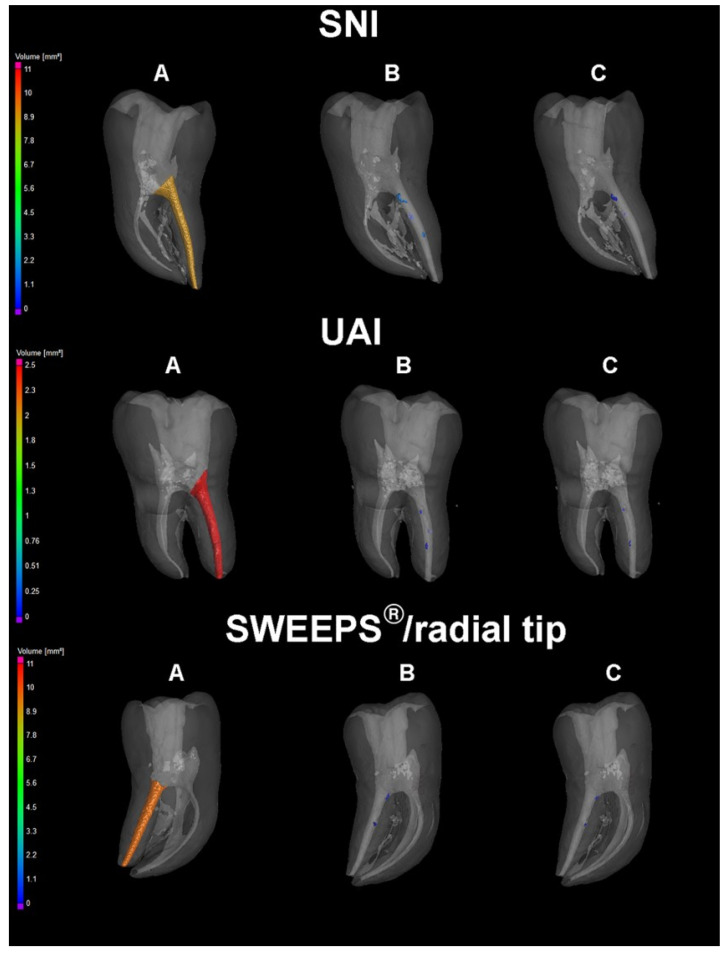
Three-dimensional micro-CT scans of the samples randomly selected from syringe-needle irrigation group (SNI), ultrasonically irrigation group (UAI) and SWEEPS group (SWEEPS) after root canal filling (**A**), basic retreatment (**B**) and final irrigation protocol (**C**).

**Table 1 bioengineering-09-00820-t001:** Presentation of the volume (in mm^3^) of the filling material initially (Volume I), after the retreatment (Volume II), and after final irrigation protocol (Volume III) and reduction rate (in %) of the volume of the filling material in each retreatment phase.

Group		Mean	SD	Minimum	Maximum	Percentiles		
						25th	50th (Median)	75th
Volume I	SNI	6.08	1.88	3.80	9.70	4.10	6.30	7.05
	SWEEPS	6.95	3.42	3.47	16.00	4.68	5.98	8.20
	UAI	6.49	2.64	3.90	12.00	4.59	5.26	8.45
Volume II	SNI	0.85	0.57	0.11	1.85	0.36	0.81	1.25
	SWEEPS	0.72	0.72	0.01	2.33	0.17	0.51	0.94
	UAI	0.89	0.80	0.05	2.89	0.32	0.63	1.46
Volume III	SNI	0.71	0.57	0.05	1.74	0.26	0.48	1.17
	SWEEPS	0.42	0.59	0.00	2.19	0.00	0.26	0.59
	UAI	0.58	0.53	0.00	1.55	0.15	0.36	1.16
Difference volume II vs. I	SNI	−86.61%	7.67%	−97.11%	−72.78%	−93.25%	−85.56%	−81.46%
	SWEEPS	−90.90%	7.05%	−99.75%	−76.74%	−96.75%	−91.89%	−86.02%
	UAI	−87.20%	9.36%	−98.88%	−69.61%	−93.59%	−90.45%	−79.43%
Difference volume III vs. I	SNI	−88.74%	8.08%	−99.16%	−74.42%	−95.38%	−89.52%	−82.75%
	SWEEPS	−95.12%	6.08%	−100.00%	−78.05%	−99.94%	−97.31%	−91.74%
	UAI	−91.15%	8.85%	−100.00%	−69.61%	−96.64%	−93.85%	−89.07%
Difference volume III vs. II	SNI	−22.89%	28.01%	−80.91%	0.00%	−42.26%	−9.75%	−2.08%
	SWEEPS	−55.98%	36.99%	−100.00%	0.00%	−99.33%	−55.05%	−26.24%
	UAI	−36.24%	32.04%	−100.00%	0.00%	−60.58%	−25.33%	−7.58%

SNI—syringe-needle irrigation. UAI—ultrasonically activated irrigation. SD—standard deviation.

## Data Availability

Not applicable.

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
