# Peer review of "The Efficacy of Er:YAG Laser-Activated Shock Wave-Enhanced Emission Photoacoustic Streaming Compared to Ultrasonically Activated Irrigation and Needle Irrigation in the Removal of Bioceramic Filling Remnants from Oval Root Canals—An Ex Vivo Study"

_bioengineering, 2022, doi:10.3390/bioengineering9120820_

Round 1
Reviewer 1 Report (Previous Reviewer 1)
This article can be accepted in this format.
Author Response
Dear reviewer,
Thank you for your helpful suggestions.
Reviewer 2 Report (Previous Reviewer 2)
The authors didn't give any modification and I still recommend to publish this manuscript as short communication.
Author Response
Dear reviewer,
Thank you for your suggestion. However, we do not agree to consider this paper as communication report. The study is of appropriate length with more than 4000 words and 34 references, covers the topic that is continuation of recent studies published in the literature. The aim of the study is clear evaluating the efficacy of a new laser activated technique and comparing it with "golden standard" in activated irrigation, ultrasonic activation, and conventional passive needle irrigation in oval root canals. The specificity and novelity of the study is that it evaluates the efficacy in complex oval anatomy which is alwals challenging in clinical pratice.
Reviewer 3 Report (New Reviewer)
This study is a mediocre in vitro study with only 3 groups and a very small sample. It is not by any means suitable for publication in a journal with an impact factor above 5. At best, it can be published in a journal without any impact factor. I suggest the authors submit this paper to "Lasers in Dental Sciences" which has no impact factor, and may be a better venue for this study. Find them at: https://www.springer.com/journal/41547
Anyways, this study is not at all appropriate for a journal with an impact factor below 1, let alone a journal with an impact factor above 5.
Author Response
Dear Reviewer,
This study covers the topic that has been investigated intensively for the last few years. It investigated the efficacy of a novel laser activated irrigation technique, SWEEPS, in the retreatment of complex oval canal anatomy. The technique was compared to “the golden standard” in activated irrigation, ultrasonic activation, and conventional needle irrigation. The justification of the aim of the study was to evaluate the benefit of the SWEEPS, which is an expensive technique, compared to very often used techniques in clinical practice. Guide in the creation of the methodology was recent review by Boutsioukis and Arias-Moliz: Present status and future directions - irrigants and irrigation methods. Int Endod J. 2022 May;55 Suppl 3(Suppl 3):588-612., in which the need to compare new techniques with standard ones is especially emphasized. The methodology of the study is novel and complex, including micro-CT scanning of the samples. Due to very detail results obtained by micro-CT and according to previous studies on the topic, each group consisted of 14 samples.
In a very recent study by Kapetanović Petričević G. et al. (Petričević GK, Katić M, Anić I, Salarić I, Vražić D, Bago I. Efficacy of different Er:YAG laser-activated photoacoustic streaming modes compared to passive ultrasonic irrigation in the retreatment of curved root canals. Clin Oral Investig. 2022 Nov;26(11):6773-6781.) (Q1, IF 3.573), SWEEPS technique was compared with PIPS technique in the retreatment of curved root canals, including 10 samples per group. In a study Jiang et al. (Jiang S, Zou T, Li D, Chang JW, Huang X, Zhang C. Effectiveness of Sonic, Ultrasonic, and Photon-Induced Photoacoustic Streaming Activation of NaOCl on Filling Material Removal Following Retreatment in Oval Canal Anatomy. Photomed Laser Surg. 2016 Jan;34(1):3-10) (IF :2.796, Q1) using same methodology (micro-CT scanning) and similar topic of research used 7 samples per group. In a study by Suk et al. (Suk M, Bago I, Katić M, Šnjarić D, Munitić MŠ, Anić I. The efficacy of photon-initiated photoacoustic streaming in the removal of calcium silicate-based filling remnants from the root canal after rotary retreatment. Lasers Med Sci. 2017 Dec;32(9):2055-2062.) (Q1, IF 3.161) used also micro-CT scanning to test thee irrigation techniques in the retreatment of filling material and included 12 samples per group. There are many studies on the similar topic evaluating the retreatment ability of irrigation of instrumentation systems in various canal anatomy, using 8-14 samples per group published in high impacted journals (Rossi-Fedele G, Ahmed HM. Assessment of Root Canal Filling Removal Effectiveness Using Micro-computed Tomography: A Systematic Review. J Endod. 2017 Apr;43(4):520-526. Q1; Keleş A, Arslan H, Kamalak A, Akçay M, Sousa-Neto MD, Versiani MA. Removal of filling materials from oval-shaped canals using laser irradiation: a micro-computed tomographic study. J Endod. 2015 Feb;41(2):219-24. Q1; Bago I, Plotino G, Katić M, Ferenac A, Petričević GK, Gabrić D, Anić I. Effect of a novel laser-initiated photoacoustic activation of a solvent or sodium hypochlorite in the removal of filling remnants after retreatment of curved root canals. Photodiagnosis Photodyn Ther. 2021 Dec;36:102535. Q2). This is obvious also from the reference list of the manuscript, which includes significant studies on the topic published in the last 6 years.
Reviewer 4 Report (New Reviewer)
Dear Authors the paper is interesting but it is necessary to review some points
-About the Title of the article,I suggest you to modify it and add the type of article.
-Please be sure to use only keywords accordingly to medical subject headings (Mesh word) for a better indexing.
- The introduction section is very short and is needed to add other references to increase the quality of the manuscript, Add recent references about the topic of the article, dwelling in the introduction on articles published in 2022.
Preferably a published articles should be with 90 or more references.
I suggest you some articles about the prosthodontic treatment post-endodontically treatment that will help you improve your article.
Telescopic overdenture on natural teeth: Prosthetic rehabilitation on OFD syndromic patient and a review on available literature PubMed ID 29460531
Prosthodontic Treatment in Patients with Temporomandibular Disorders and Orofacial Pain and/or Bruxism: A Review of the Literature https://doi.org/ 10.3390/prosthesis4020025
-I suggest you add a table with the list of abbreviations used in the text.
Thank You,
Kind Regards
Author Response
-About the Title of the article,I suggest you to modify it and add the type of article.
Thank you for your suggestion. The title was changed according to your suggestion.
-Please be sure to use only keywords accordingly to medical subject headings (Mesh word) for a better indexing.
Thank you for your suggestion.
- The introduction section is very short and is needed to add other references to increase the quality of the manuscript, Add recent references about the topic of the article, dwelling in the introduction on articles published in 2022.
New references have been added in Introduction.
Preferably a published articles should be with 90 or more references.
Thank you for your comment. We consider the manuscript cover the most recent referencess on the topic and regarding it is an original science paper, not a review, 39 references supposed to be enough to explain and present the results of the study.
I suggest you some articles about the prosthodontic treatment post-endodontically treatment that will help you improve your article.
Telescopic overdenture on natural teeth: Prosthetic rehabilitation on OFD syndromic patient and a review on available literature PubMed ID 29460531
Prosthodontic Treatment in Patients with Temporomandibular Disorders and Orofacial Pain and/or Bruxism: A Review of the Literature https://doi.org/ 10.3390/prosthesis4020025
The manuscript covers the topic on the retreatment procedures and evaluated the efficacy of different irrigation techniques in the retreatment of bioceramic sealer. Therefore, we consider that including studies on post-endodontic restorations would possibly confuse the readers.
-I suggest you add a table with the list of abbreviations used in the text.
The list of abbreviations was added in the text.
Reviewer 5 Report (New Reviewer)
Thank you for your research and submission regarding the paper “Evaluation of the efficacy of Er:YAG laser activated shock-wave enhanced emission photoacoustic streaming compared to ultra-sonically activated irrigation and needle irrigation in the removal of bio ceramic filling remnants from oval root canals”.
Studies with small samples (n=42, 3 groups of 14) might seem small in order to obtain significant results, but even when non-significant results are obtained it’s important to publish in order to leave them ready to enter systematic and meta-analysis studies (therefore it´s important to leave the statistical results complete as you did, presenting both the median values that you use for comparing purposes as well as the average and standard deviation that might be used in later meta-analysis).
In the material and methods section (or any other location) nothing is said about the Ethics Committee information regarding this study, as it uses human extracted teeth. Did you obtain authorization? If not, is there a reason for not needing one?
Result section: Line 202-203 there is a paragraph mistake (not needed).
Although you use p-values for significant results or non-significant using p<0.05/p<0.001 and p>0.05, respectively, it would be extremely interesting to see the real p-values (you should include them in table 1). P-values above 0.05 are non-significant, but when having very small sample/group sizes it is different to have a p=0.097 or a p=0.697. The thing is: did you obtain p-values that might indicate a type II error?
Interesting discussion but another limitation should be added, regarding the small sample size used.
Author Response
Studies with small samples (n=42, 3 groups of 14) might seem small in order to obtain significant results, but even when non-significant results are obtained it’s important to publish in order to leave them ready to enter systematic and meta-analysis studies (therefore it´s important to leave the statistical results complete as you did, presenting both the median values that you use for comparing purposes as well as the average and standard deviation that might be used in later meta-analysis).
Thank you for your comment. The methodology used in the study is micro-CT scanning which provide very detail results regarding the volume of the material left inside the small areas root cana. Accoridng to previous studies on the similar topic, also using micro-CT scanning, 7-12 samples per group were used (Petričević GK, Katić M, Anić I, Salarić I, Vražić D, Bago I. Efficacy of different Er:YAG laser-activated photoacoustic streaming modes compared to passive ultrasonic irrigation in the retreatment of curved root canals. Clin Oral Investig. 2022 Nov;26(11):6773-6781; Bago I, Plotino G, Katić M, Ferenac A, Petričević GK, Gabrić D, Anić I. Effect of a novel laser-initiated photoacoustic activation of a solvent or sodium hypochlorite in the removal of filling remnants after retreatment of curved root canals. Photodiagnosis Photodyn Ther. 2021 Dec;36:102535.; Jiang S, Zou T, Li D, Chang JW, Huang X, Zhang C. Effectiveness of Sonic, Ultrasonic, and Photon-Induced Photoacoustic Streaming Activation of NaOCl on Filling Material Removal Following Retreatment in Oval Canal Anatomy. Photomed Laser Surg. 2016 Jan;34(1):3-10.; Rossi-Fedele G, Ahmed HM. Assessment of Root Canal Filling Removal Effectiveness Using Micro-computed Tomography: A Systematic Review. J Endod. 2017 Apr;43(4):520-526.; Keleş A, Arslan H, Kamalak A, Akçay M, Sousa-Neto MD, Versiani MA. Removal of filling materials from oval-shaped canals using laser irradiation: a micro-computed tomographic study. J Endod. 2015 Feb;41(2):219-24.)
In the material and methods section (or any other location) nothing is said about the Ethics Committee information regarding this study, as it uses human extracted teeth. Did you obtain authorization? If not, is there a reason for not needing one?
Thank you for your comment. The study is a part of big research project approved by the Local Ethics Committee with reference number 05-PA-30-VI-4/2019. This was added in the text.
Result section: Line 202-203 there is a paragraph mistake (not needed).
The section was corrected.
Although you use p-values for significant results or non-significant using p<0.05/p<0.001 and p>0.05, respectively, it would be extremely interesting to see the real p-values (you should include them in table 1). P-values above 0.05 are non-significant, but when having very small sample/group sizes it is different to have a p=0.097 or a p=0.697. The thing is: did you obtain p-values that might indicate a type II error?
Thank you for your question.
When looking the differences between the groups in percentage reduction, no significant differences were found, and therefore, all three techniques reduced the volumes equally. The closest to a significant difference were the differences in the percentage reduction of volume III compared to volume I, where the P value was 0.067. The Kruskal-Wallis test was used to assess these differences because the sample size was small.
Interesting discussion but another limitation should be added, regarding the small sample size used.
Thank you for your comment. The explanation was added in the text.
Round 2
Reviewer 2 Report (Previous Reviewer 2)
The authors have revised the manuscript reasonably. Now we recommend that the manuscript can be published in the journal of “Bioengineering”. However, a few of grammatical errors should be revised again before publication.
1. The sentence “Furthermore, the differences between groups, considering Volume III vs Volume I, were not significant, however showed values very close the level of significance (p=0.067). Therefore, future study should include bigger sample size in order to check if the difference remain the same.” should be improved.
2. The format of reference (Refs. 1, 16, 30, 37…) is not consistent. Please check again.

Reviewer 3 Report (New Reviewer)
Sorry but I still believe strongly this study is not suitable for this high-impact journal. At best, this study should be submitted to a journal with an impact factor below 1 (or even without any impact factor).
Reviewer 4 Report (New Reviewer)
Dear author,
i'm satisfied, in my opinion the paper is ready for the publication
regards
Reviewer 5 Report (New Reviewer)
Thank you for all answers and manuscript alterations.
This manuscript is a resubmission of an earlier submission. The following is a list of the peer review reports and author responses from that submission.
Round 1
Reviewer 1 Report
In methods, the power of 0.6 w is not correct. if you have 15 hz frequency and 20 mL energy, the power is different from what you mentioned.
What is the rest time between each irradiation in SWEEPS tecyhnique?
What is the pulse duration in SWEEPS technique?
The discussion can be improved by adding some new sweeps articles such as
1) Ensafi F, Fazlyab M, Chiniforush N, Akhavan H. Comparative effects of SWEEPS technique and antimicrobial photodynamic therapy by using curcumin and nano-curcumin on Enterococcus faecalis biofilm in root canal treatment. Photodiagnosis Photodyn Ther. 2022 Sep 24;40:103130. doi: 10.1016/j.pdpdt.2022.103130. Epub ahead of print. PMID: 36162755.
2) Koruk D, Basmacı F, Kırmızı D, Aksoy U. The Impact of Laser-Activated and Conventional Irrigation Techniques on Sealer Penetration into Dentinal Tubules. Photobiomodul Photomed Laser Surg. 2022 Aug;40(8):565-572. doi: 10.1089/photob.2022.0017. Epub 2022 Aug 2. PMID: 35917520.
Author Response
Answers to the reviewer's comments 1
- In methods, the power of 0.6 w is not correct. if you have 15 hz frequency and 20 mL energy, the power is different from what you mentioned.
Thank you for your comment. You are right. Based on the 15hz frequency and 20 mJ pulse energy, the average power is 0.3 W. The pulse duration is 25 µs. It was corrected.
- What is the rest time between each irradiation in SWEEPS tecyhnique?
Thank you for your question. In this study, the SWEEPS technique was used in combination with only 6 mL of 3% NaOCl, used in three cycles of 30 s . That is total of 90 s of activated irrigation. The purpose of that irrigation was to further enhance the removal of bioceramic filling remnants from root canals. The protocol was according to previous studies on the same topic: Petričević GK, Katić M, Anić I, Salarić I, Vražić D, Bago I. Efficacy of different Er:YAG laser-activated photoacoustic streaming modes compared to passive ultrasonic irrigation in the retreatment of curved root canals. Clin Oral Investig. 2022 Jul 26. doi: 10.1007/s00784-022-04637-0. Epub ahead of print.
- What is the pulse duration in SWEEPS technique?
Thank you for your comment. The pulse duration was mentioned in Introduction (Page 2, Line 56) and is 25 µs. This short pulse duration is specifically for SWEEPS technique causing extremely high peak power.
- The discussion can be improved by adding some new sweeps articles such as
1) Ensafi F, Fazlyab M, Chiniforush N, Akhavan H. Comparative effects of SWEEPS technique and antimicrobial photodynamic therapy by using curcumin and nano-curcumin on Enterococcus faecalis biofilm in root canal treatment. Photodiagnosis Photodyn Ther. 2022 Sep 24;40:103130. doi: 10.1016/j.pdpdt.2022.103130. Epub ahead of print. PMID: 36162755.
2) Koruk D, Basmacı F, Kırmızı D, Aksoy U. The Impact of Laser-Activated and Conventional Irrigation Techniques on Sealer Penetration into Dentinal Tubules. Photobiomodul Photomed Laser Surg. 2022 Aug;40(8):565-572. doi: 10.1089/photob.2022.0017. Epub 2022 Aug 2. PMID: 35917520.
Thank you for your suggestion. The articles were added in Discussion section.
Reviewer 2 Report
This work experimentally compared three activated techniques, SWEEPS, UAI, and SNI to remove the BioRoot RCS filling remnants from oval root canals after chemo-mechanical instrumentation using a reciprocating single file. All three final irrigation protocols contributed to the removal of BC/gutta-percha filling remnants without significant differences between them. However, the SWEEPS mode has great potential for complete removal of filling remnants during the retreatment, and this conclusion is not so strong.
Although the research is designed appropriate, the methods are adequately described, and the results are clearly presented, the physics or mechanism is not sufficient and the advantage of the new method is not well desmonstrated. In this sense, I suggest to publish this manucript as a short communication.
Author Response
Dear reviewer,
Thank you for your comment. The aim of this article was to compare the new method of laser activated irrigation, the SWEEPS technique, in the removal of filling remnants from oval distal root canal from molars. It is a continuation of the previous study on the same topic: Petričević GK, Katić M, Anić I, Salarić I, Vražić D, Bago I. Efficacy of different Er:YAG laser-activated photoacoustic streaming modes compared to passive ultrasonic irrigation in the retreatment of curved root canals. Clin Oral Investig. 2022 Jul 26. doi: 10.1007/s00784-022-04637-0. Epub ahead of print. . In that study, the SWEEPS technique was not better than ultrasonic irrigation in the removal of the filling remanants from severly curved root canals. These similar results pointed out that the choice between SWEEPS and LAI is not mandatory when dealing with oval and curved canals. Therefore, I think this study should be published as an original science paper.
Reviewer 3 Report
P2 line 80: A minimum of 10 canals was calculated as sample size for each group
P2 line 82: A total of 42 oval distal root canals with straight curvature (<5%) were selected for the study.
It is not clear how many teeth used in this experiment.
The demographic profile of used teeth needs to be addressed.
Did this experiment obtain form the IRB?
It is not clear the repeated number of each micro-CT scanning and software measurement.
Author Response
P2 line 80: A minimum of 10 canals was calculated as sample size for each group
P2 line 82: A total of 42 oval distal root canals with straight curvature (<5%) were selected for the study.
It is not clear how many teeth used in this experiment.
Thank you for your comment. The statistical analysis showed that minimum of 10 samples would be enough for the study. However, based on our previous studies on the topic showing problems regarding differences in canal anatomy, we decided to increase the number of samples in each group. So, finally we manage to include 14 samples with similar anatomy per group, based on CBCT. Micro-CT analysis also showed good distribution of samples per group and there was no need for excluding some samples from the study. This was clarify in the study.
The demographic profile of used teeth needs to be addressed.
Thank you for your suggestion. The teeth were selected from patients aged between 22 and 42 years old.
Did this experiment obtain form the IRB?
This study was conducted within the research project of the Croatian Science Foundation.
It is not clear the repeated number of each micro-CT scanning and software measurement.
Each tooth sample was scanned in microCT for three times: initially after root canal filling, after basic retreatment and after additional irrigation protocol using on of the tested technique (SWEEPS, UAI or SNI).
All the samples were scanned at the same position with the same radiation settings. With the gray scale value for the tooth as the base value, a simple threshold algorithm was used to detect the volume of filling material in the internal tooth volume. The results were expressed as a percentage of the remaining filling material with respect to the initial volume of the root canal filling by using relational values. The variations in sample volumes were effectively excluded from the analysis of the material removal rate. The same procedure was applied for all samples, thus providing a constant metric for the rate of removal of material in the root canal.